# The Chemical Composition, Fermentation End-Product of Silage, and Aerobic Stability of Cassava Pulp Fermented with *Lactobacillus casei* TH14 and Additives

**DOI:** 10.3390/vetsci9110617

**Published:** 2022-11-07

**Authors:** Sunisa Pongsub, Chanon Suntara, Waroon Khota, Waewaree Boontiam, Anusorn Cherdthong

**Affiliations:** 1Tropical Feed Resources Research and Development Center (TROFREC), Department of Animal Science, Faculty of Agriculture, Khon Kaen University, Khon Kaen 40002, Thailand; 2Department of Animal Science, Faculty of Natural Resources, Rajamangala University of Technology Isan, Sakon Nakhon Campus, Phangkhon, Sakon Nakhon 47160, Thailand

**Keywords:** cassava pulp, *Lactobacillus casei* TH14, urea, molasses, fermentation

## Abstract

**Simple Summary:**

Ruminant feed is one potential application for cassava pulp (CSP), an industrial waste product generated during the manufacturing of cassava flour. However, because of its high fiber content and low protein content, the quality should be enhanced. We observed that fermenting CSP with urea, molasses, and *Lactobacillus casei* TH14 improves silage quality, which might be utilized as an alternative feed source for ruminants.

**Abstract:**

This study evaluated the effects of cassava pulp fermented with *Lactobacillus casei* TH14, urea, and molasses on its chemical composition, the fermentation end-product of silage, and aerobic stability. A 2 × 2 × 2 factorial arrangement with a randomized complete block design was employed. The first factor: level of *L. casei* TH14 [L; 0 and 10^5^ cfu/kg fresh matter (FM)], the second factor: level of molasses (M; 0 and 4% DM), the third factor: level of urea (U; 0 and 4% DM), and the number of days of fermentation (7, 14, and 21 days) were evaluated using a statistical block. There were interactions among CSP fermented with different additives on DM content (*p* < 0.05). The control group (CON) and CSP fermented with L, L×M, and L×U had lower DM contents than U, U×M, and L×U×M. The crude protein of CSP was increased by interaction of L×U and U×M additives (*p* < 0.05 and *p* < 0.01, respectively). Interaction effects between L and U and NDF content were detected (*p* < 0.05). The L×U combination resulted in a significantly lower NDF than the other groups. The interaction between L×U×M had no effect on the change in the CSP fermentation process (*p* > 0.05). The combination of U×M caused a poorer pH than other groups (*p* < 0.01). The ammonia-N content was higher than others, when CSP was fermented with L×U (*p* < 0.01) or U×M (*p* < 0.05), respectively. The lactic acid levels in fermented CSP were higher (*p* < 0.01) than in other groups through the *L. casei*. The interaction between L×U×M had an influence on lactic acid bacteria (LAB) (*p* < 0.01) and aerobic bacteria (*p* < 0.01). The highest LAB population (*p* < 0.01) at 10^6^ cfu/g FM was found in CSP fermented with *L. casei* and molasses. In conclusion, the current study shows that CSP treated with L×U×M resulted in good preservation by recovering DM, a low number of aerobic bacteria, and greater LAB than other treatments, with the exception of the L×U×M addition. A 21-day fermentation period is advised because it produces products with greater levels of crude protein, lactic acid, acetic acid, and propionic acid.

## 1. Introduction

Cassava (*Manihot esculenta* Crantz), which ranks third in economic significance in Thailand behind rice and rubber, is a major crop. Global production of cassava was 277 million tons in 2018, and 302 million tons are expected to be produced in the crop year of 2021/22, an increase of 9.0% [1,2,3]. The top five cassava-producing countries in the world are Nigeria (20.37%), Congo (10.83%), Thailand (10.61%), Indonesia (6.52%), and Brazil (6.47%) [3,4]. The waste and residues from Thailand’s cassava starch production amount to 7.4 million tons of liquid waste yearly and 3.4 [5] and 7.3 [6] million tons per year of cassava pulp (CSP). CSP is a byproduct of industry, generated during the manufacturing of cassava flour. CSP decomposes quickly, causing an environmental risk, a strong and terrible decomposition smell, and local water contamination [7]. Nevertheless, the CSP maintains its nutritional content, which increases its demand in the feed industry. Additionally, CSP material also contains considerable amounts of carbohydrates, which ruminants can use as a source of energy [8]. However, previous research discovered problems when it was utilized as ruminant feed. According to Kongphitee et al. [9], the inclusion of 50% CSP in the total mixed ration (TMR) could reduce the intake of acid detergent fiber (ADF) by 54% in Thai native beef cattle. Additionally, Keaokliang et al. [10] illustrated that Thai native beef cattle fed with a TMR diet (9.6% crude protein; CP) containing 29.8% CSP could reduce protein digestion, urinary nitrogen, retained nitrogen, and the urine-to-gross-energy ratio in comparison to the control group, which was fed a TMR diet with 12.6% crude protein and no CSP. Applying different techniques has been suggested to increase the usage of CSP in animal feed.

Many studies have been interested in improving the quality of CSP in animal diets through physical, chemical, and physicochemical means [11]. CSP is currently treated with various biological and chemical additives such as *Saccharomyces cerevisiae* [12], replacing soybean meal with CSP fermented with yeast waste [13], *Lactobacillus casei* TH14 (LcTH14), molasses, and cellulase [14] to improve its nutritive value and make it an appropriate choice for animal diets. Additionally, several studies have investigated how *S. cerevisiae* affects animal diets and how it improves the quality of byproducts, such as CSP, and they recommend feeding animals [13,14,15]. This could be because *S. cerevisiae* has a high protein content and is rich in the essential amino acids animals require. Phesatcha et al. [15] have shown that adding 15 g/d of *S. cerevisiae* to fattening cattle enhanced nutrient digestibility and growth performance. Additionally, the capability of yeast cells to scavenge oxygen may be one of the primary factors that justify the beneficial effect that live yeasts have on the quality of silage as well as fiber-degrading bacteria in the rumen [12,13]. In a similar fashion, Suntara et al. [12] found that certain strains of yeast were able to produce cellulolytic enzymes, which were capable of degrading the fibers present in plant materials. Regarding the use of LAB to improve silage quality, Pholsen et al. [16] compared homofermentative types of LAB, including *Lactobacillus casei* strain TH14, Lactobacillus plantarum for three strains, and *Lactobacillus rhamnosus*. They discovered that strain TH14 has a high lactic-acid-production capacity and can produce more lactic acid, which verified its usefulness as a possible silage inoculant and showed that this strain was more successful in enhancing silage quality than inoculants or other strains.

This is due to the fact that adding urea and molasses as cheap and accessible silage additives uses them as sources of NPN and carbohydrates for silage additives, respectively. Lactic acid bacteria inoculation aids in successful fermentation processes. Furthermore, urea was typically added to low nitrogen feedstuffs, and the current cassava pulp contained a low quantity of nitrogen. A source of nitrogen can also be used in the silage as a substrate for the growth of microorganisms [14]. A previous study successfully improved the quality of low nitrogen roughages by fermenting them with a mixture of *L. casei* TH14, urea, and molasses. Cherdthong et al. [17] discovered that ensiled rice straw with 50 g molasses/kg fresh matter (FM) plus *L. casei* TH14 (10^5^ cfu/kg FM of rice straw) provides better results, by preventing CP loss during fermentation while also increasing beef cattle digestibility, rumen bacterial population, and propionic acid. Similarly, sugarcane bagasse fermented with *L. casei* at 10^5^ cfu/kg and 50 g/kg molasses can increase bagasse quality while also improving feed utilization, rumen ecology, and milk production in Holstein Friesian cows [18]. The addition of *L. casei* TH14 and cellulase increased CP and decreased neutral detergent fiber in sweet corn stover silage, according to Kaewpila et al. [14]. However, there have been no studies performed on how to use LcTH14 combined with urea and molasses for animal feed to enhance the quality of CSP fermentation. The purpose of this study was to evaluate the effects of fermented cassava pulp (CSP) with *L. casei* TH14 (LcTH14), urea, and molasses on chemical composition, the fermentation end-product of silage, microbial population, and aerobic stability. 

## 2. Materials and Methods

### 2.1. Treatments and Experiment Design

In this experiment, a randomized complete block design with a 2 × 2 × 2 factorial arrangement was used, wherein the first factor: the level of LcTH14 (0 and 1 × 10^5^ colony forming unit; cfu/kg FM) [19], the second factor: the level of urea (0 and 4% DM) [20], the third factor: the level of molasses (0 and 4% DM) [20], and block: the period of fermentation (7, 14, and 21 days). The treatment combinations of additives for fermented CSP are described below:
(1)No LcTH14 + Urea 0% DM + Molasses 0% DM; CON(2)No LcTH14 + Urea 0% DM + Molasses 4% DM; M(3)No LcTH14 + Urea 4% DM + Molasses 0% DM; U(4)No LcTH14 + Urea 4% DM + Molasses 4% DM; UM(5)LcTH14 1 × 10^5^ cfu/g FM + Urea 0% DM + Molasses 0% DM; L(6)LcTH14 1 × 10^5^ cfu/g FM + Urea 0% DM + Molasses 4% DM; LM(7)LcTH14 1 × 10^5^ cfu/g FM + Urea 4% DM + Molasses 0% DM; LU(8)LcTH14 1 × 10^5^ cfu/g FM + Urea 4% DM + Molasses 4% DM; LUM

### 2.2. Material and Silage Preparation

The CSP (wet form) material was obtained in Khon Kaen, northern Thailand. The CSP had fermented while being kept and covered by a tarp for a few days for the experiment. The recommended LcTH14 was purchased, containing 10^11^ cfu/g LcTH14 (Bio Ag Khon Kaen Co., Ltd., Khon Kaen, Thailand) [21]. On the packaging, it was advised to use 50 g of LcTH14 for every 10 tons of material. Urea was used in the form of granular agricultural grade urea 46% nitrogen fertilizer (urea 46-0-0 fertilizer), and molasses was purchased from the Mitr Phol molasses brand, produced by the Mitrphol sugar producer in Khon Kaen province. Each LcTH14 was individually dissolved in distilled water that had been sterilized. Molasses and urea (4% DM consumption rate) [20] were both dissolved in sterile distilled water. For laboratory-scale experiments, 5000 g of the materials were separated into 350-g bags and vacuum-sealed by Brother (Zhejiang Brother Packing Machinery Co., Ltd., Wenzhou, Zhejiang, China) at room temperature. The three replications were prepared for 7, 14, and 21 days of fermentation. The silage was separated from the silo, mixed well, and kept at 10 g for an analysis of the pH value and microbial population, 20–30 g for dry matter content, 150 g for chemical composition, and 150 g for aerobic stability.

### 2.3. Chemical Composition 

The samples were dried in an oven at 60 °C for each replication of each treatment, then mashed into smaller particles using a 1-mm colander screen to measure the chemical compositions, including dry matter (DM-viz. 34.01), organic matter (OM-viz. 942.05), crude protein (CP-viz. 976.05), and ether extract (EE-viz. 920.39), following the standard method of the Association of Official Analytical Chemists [22]. The neutral detergent fiber (NDF) and acid detergent fiber (ADF) were evaluated using the method of Van Soest et al. [23] and followed by evaluation with the use of an Ankom fiber analyzer (ANKOM 200, ANKOM Technology, New York, NY, USA). 

### 2.4. Fermentation End-Product Analysis of Silage 

At the end of fermentation, a blend of 10 g of sample and 90 mL of sterilized distilled water was placed in an incubator for 12 h at 4 °C [24]. The pH meter was then used to measure the pH value (Analog pH/mV/°C meter with HI1230B electrode—HI83141, HANNA Instruments, Inc., Woonsocket, RI, USA). The levels of lactic acid (LA) and volatile fatty acids (acetic acid, AA; propionic acid, PA; butyric acid, BA) were measured using a gas chromatograph (Nexis GC-2030: SHIMADZU, Shimadzu Corp., Kyoto, Japan) equipped with a capillary column (molecular sieve 13X, 30/60 mesh, Alltech Associates Inc., Deerfield, IL, USA) [25]. The ammonia nitrogen (NH_3_-N) concentration was evaluated by a spectrophotometer (UV/Vis Spectrometer, PG Instruments Ltd., London, UK) based on the method of Fawcett and Scott [26].

### 2.5. Microbial Counting

The silage samples for microbial analysis were obtained in a vacuum bag and stored at room temperature (28–30 degrees Celsius). They were collected at four different points of the silo; point 1: top left side; point 2: top right sides; point 3: intermediate regions between the center and left side at the bottom of the silo; point 4: intermediate regions between the center and right side at the bottom of the silo; and these subsamples were manually collected and homogenized, forming one composite sample per silo, following the methods of Carvslho et al. [27]. The samples of CSP were evaluated for LAB, coliform bacteria, aerobic bacteria (AB), yeast, and mold populations using a plate-counting method [14,28] based on the original method developed by Kozaki et al. [29]. LAB was evaluated on MRS agar (Difco Laboratories, Detroit, MI, USA) after 48 h of incubation in an anaerobic jar (Sugiyamagen Ltd., Tokyo, Japan). Coliform bacteria were evaluated on blue-light broth agar (Nissui-seiyaku Ltd., Tokyo, Japan) after 2 d of incubation. After 24 h of incubation at 30 °C, the aerobic bacteria were evaluated on nutrient agar (Difco Laboratories Inc.). After incubating at 30 °C for 24 h, yeast and mold were tested on potato dextrose agar (Nissui Ltd., Tokyo, Japan). By examining the morphology of the cells and the colony structure, yeasts were distinguished from molds or bacteria.

### 2.6. Aerobic Stability Analysis

On each of the fermentation days for all silages, the aerobic stability was assessed. A total of 350 g of each treatment’s three duplicates were kept loosely packed in silos and allowed to degrade aerobically at the ambient temperature (27–34 °C). Each container was covered with two layers of cheesecloth to prevent drying and contamination while allowing air to flow through [30]. Aerobic stability was measured using digital thermometer readings once every 6 h in 3 replicates. Thermocouple wires were attached to the silage. Aerobic deterioration was estimated by days (or hours) until the start of a continued rise in temperature by more than 2 °C above the ambient temperature [31].

### 2.7. Statistical Analysis 

The chemical composition, pH values, and fermentation characteristics of CSP after fermentation were measured in a 2 × 2 × 2 factorial in a randomized complete block design by the effect of two levels of LcTH14, two levels of urea, and two levels of molasses, and the block was three days of fermentation. We used SAS version 9.00 (SAS Institute Inc., Cary, NC, USA) with the following model:$$Y_{ijkl}=µ+\rho_{i}+\alpha_{j}+\beta_{k}+\gamma_{l}+\alpha \beta_{jk}+\alpha \gamma_{jl}+\beta \gamma_{kl}+\alpha \beta \gamma_{jkl}+\varepsilon_{ijkl}$$
where *Y_ijkl_* = observation values; µ = overall mean; *i* = effect of block at I when *i* = 1, …, r; *α_j_* = effect of main effect A at *j* when *j* = 1, …, a; *β_k_* = effect of main effect B at *k* when *k* = 1, …, b; *γ_l_* = effect of main effect C at *l* when *l* = 1, …, c; *αβ_jk_* = interaction of A and B at *jk*; *αγ_jl_* = interaction of A and C at *jl*; *βγ_kl_* = interaction of B and C at *kl*; *αβγ_jkl_* = interaction of A, B, and C at *jkl*; *ε_ijkl_* = error term. The block was three days of fermentation. 

The aerobic stability and microbial population after 21 days of fermentation were analyzed with SAS version 9.00 (SAS Institute Inc., Cary, NC, USA) with the following model:$$Y_{ijk}=µ+\alpha_{i}+\beta_{j}+\gamma_{k}+\alpha \beta_{ij}+\alpha \gamma_{ik}+\beta \gamma_{jk}+\alpha \beta \gamma_{ijk}+\varepsilon_{ijk}$$
where *Y_ijk_* = observation values; µ = overall mean; *α_i_* = effect of main effect A at *i* when *i* = 1, …, a; *β_j_* = effect of main effect B at *j* when *j* = 1, …, b; *γ_k_* = effect of main effect C at *k* when *k* = 1, …, c; *αβ_ij_* = interaction of A and B at *ij*; *αγ_ik_* = interaction of A and C at *ik*; *βγ_jk_* = interaction of B and C at *jk*; *αβγ_ijk_* = interaction of A, B, and C at *ijk*; *ε_ijk_* = error term. 

Each treatment had three replications. Using Duncan’s new multiple range tests [32], the means were compared at *p* < 0.05, which is accepted as a statistically significant difference.

## 3. Results

### 3.1. Microbiology and Chemical Composition of Cassava Pulp before Fermentation

The microbial populations and chemical composition of CSP before ensiling are shown in Table 1. The CSP had a pH of 3.96 before it was ensiled, and neither coliform bacteria nor molds were found in it.

### 3.2. Chemical Composition of Cassava Pulp Treated with LcTH14, Urea, and Molasses after Fermentation

The chemical composition of CSP fermented with various additions is presented in Table 2. The fermentation of CSP with L × U × M had no effect on OM or ADF (*p* > 0.05). However, there were interactions among CSPs fermented with different additives on DM content (*p* < 0.05). The control group and CSP fermented with L, L×M, and L × U had lower DM contents than U, U × M, and L × U × M. CP was higher by the interaction of L × U and U × M (*p* < 0.05 and *p* < 0.01, respectively). As a result of a single factor from U supplementation, CS’s EE decreased (*p* < 0.05). Interaction effects between L and U and NDF content were detected (*p* < 0.05). The L × U combination resulted in a significantly lower NDF than the other groups. The influence of the day of fermentation (block) on the EE and ADF contents in CSP had no effect (*p* > 0.05). However, the largest CP (*p* < 0.01) was obtained after 21 days of CSP fermentation (Figure A1B), but fermentation processes lasting longer than 7 days reduced the OM and NDF values ((*p* < 0.05) and (*p* < 0.01), respectively) (Figure A1C,D). Furthermore, CSP fermentation lasting 7 days had a higher DM content than that lasting 14 and 21 days (Figure A1A). 

### 3.3. Effect of Cassava Pulp Treated with LcTH14, Urea, and Molasses on Characteristics after Fermentation

The characteristics of the CSP fermentation product are shown in Table 3. The interaction between L × U × M had no effect on the change in the CSP fermentation process (*p* > 0.05). The combination of U × M caused a pH that was lower than in other groups (*p* < 0.01). The PA and ammonia-N contents were higher than others when CSP was fermented with L × U (*p* < 0.01, *p* < 0.05) or U × M (*p* < 001, *p* < 0.05), respectively. LA concentrations in fermented cassava pulp were higher with LcTH14 addition than without (*p* < 0.01). The day of fermentation of CSP had no effect on the ammonia-N level (*p* > 0.05). The amount of LA produced was higher when CSP fermentation had been carried out for 7 days (*p* < 0.01) (Figure A2B), whereas the pH value and the level of organic acids (AA, PA, and BA) were higher when fermented for more than 14 days (*p* < 0.01, *p* < 0.05, *p* < 0.05, and *p* < 0.05, respectively) (Figure A2A,C–E). The pH values for CSP fermentations that occurred over a period of more than 14 days were higher after 7 days (*p* < 0.01).

### 3.4. Microbiology and Aerobic Stability of Cassava Pulp Treated with LcTH14, Urea, and Molasses after 21 Days of Fermentation

The effects of LcTH14, urea, and molasses on microbial populations and aerobic stability after 21 days of fermentation are shown in Table 4. Aerobic stability was stable for more than 120 h after opening the silo for all additives. Coliform and mold populations were not detected (10^2^ cfu/g FM) during CSP fermentation in any additive. The interaction between L × U×M had no effect on yeast populations in fermented CSP (*p* > 0.05), but it has an influence on LAB (*p* < 0.01) and AB (*p* < 0.01). The highest LAB population (*p* < 0.01) at 9.1 × 10^6^ cfu/g FM was found in CSP treated with LcTH14 and molasses together, while CSP treated with additives reduced the AB population (*p* < 0.05) compared with the CON group.

## 4. Discussion

### 4.1. Microbial Populations and Chemical Composition of Cassava Pulp

In the present study, prior to improvement, the CSP contained LAB, AB, and yeasts. These findings demonstrated that LAB numbers were lower than AB and yeast, indicating that it was necessary to prevent the growth of undesirable bacteria. Additionally, a high population of yeast and AB was found in unfermented cassava pulp [33], which may be due to their ability to utilize the carbohydrate that exists as a partial substrate for microbe growth [34]. Thus, LAB and additives introduced to fermented cassava pulp might reduce yeast and AB populations. As Keawpila et al. [14] recommended, CSP is not sufficiently homofermentative-LAB biased to be able to preserve CSP. 

In agreement with the findings of this study, Keaokliang et al. [10] have found a chemical composition in which the samples were taken from various manufactories in northeastern Thailand during each season. Despite the high starch and cellulose content and low cost, CSP has been shown to be a beneficial energy source for animal feed. It has been shown that fermented CSP increases CP by 12.7% to 26.4% [12,13,20], which ultimately results in an increase in the nutritional content of animal diets [6]. However, CSP’s utilization is limited by its characteristics, including high moisture and low CP concentration [10]. In this trial, the pH value of CSP before fermentation averaged 3.96 lower than previously reported, with an average of 4.3 and 4.99, respectively [10,35]. That would be the case because the CSP samples in this experiment had fermented while being stored and covered by a tarp for the experiment for a few days. Huang et al. [36] determined that the pH of each batch of silage was significantly reduced (under 4.2 after a few days of ensiling) throughout fermentation with no additions. Moreover, sulfur dioxide (SO_2_) was also added through processing as a bleaching and antibacterial agent, which further contributed to the low pH of CSP. Therefore, the low pH of CSP is beneficial because it assists in the long-term preservation of the pulp [10,37].

### 4.2. Chemical Composition of Cassava Pulp Fermented with LcTH14, Urea, and Molasses after Fermentation

As shown in a previous study, Yunus et al. [38] found that as molasses levels increased, silage DM enhanced as well. According to Alli et al. [39], adding molasses to chopped whole-plant leucaena enhanced LA production and decreased DM losses compared to the control. Using LAB in starter cultures can improve feed quality while reducing nutrient loss, improving the fermentation process, and results in reduced DM loss and pathogenic activity [40]. The mechanism of LAB could help to inhibit the growth of undesirable microbes, reducing proteolysis and DM loss in early fermentation [40]. In addition, Santos et al. [41] observed that the addition of urea to sorghum silage reduces nutrient losses (DM) and promotes nutrient content, improving the CP content of the silage. As a result, the DM values for the CSP treated with U, U × M, and L × U × M were higher than the CON group. A possible cause could be that it helps prevent the undesirable microorganisms that lead to nutritional loss. This study’s DM content is consistent with the findings of So et al. [21], who discovered that fermenting sugarcane bagasse with *Lactobacillus casei* TH14 combined with cellulase, and molasses did not reduce DM loss. In addition, Cherdthong et al. [17] found no difference in DM loss after ensiled rice straw was treated with *Lactobacillus casei* TH14, molasses, and cellulase enzymes.

The fermentation days were lengthened in this study, which may result in lower DM and OM levels. Longer fermentation periods result in reduced OM in silage, which is consistent with Bial et al. [42]’s research. The WSC was used by LAB and other microbes for fermentation, resulting in a reduced DM level in each group as fermentation duration increased [43]. Moreover, Mayne and Gardon [44] found OM losses during the in-silo period followed a similar trend that was observed for DM losses, in which OM is progressively degraded by a complex pathway in several microbial communities under anaerobic conditions.

The CP content of the treated CSP was increased due to urea supplementation. Urea is a low-cost protein per unit, with about 46% nitrogen, which has the main objective of increasing the CP content of the feed [45]. The ammonia dissolved in the water is only weakly hydrolyzed to release hydroxyl ions in this equilibrium process (OH^−^). In a hydrolysis reaction, the hydroxyl ion produced by water dissociation combines with fats to produce free fatty acids, which bond to the ammonium ion to form soluble salts [46]. The decrease in the EE of CSP may be due to this. Fernandes et al. [47] reported a lower NDF content in sorghum silages with the addition of 2.5%, 5.0%, and 7.5% urea. The hydrolysis of urea causes certain molecules, such as cellulose, to expand and form intermolecular bonds on the cell wall [48]. As a result of ammonia’s positive relationship with water, which promotes cell-wall expansion, ammoniated forage benefits from tissue breakdown [49]. In the same way, supplementing rice silage with urea and molasses can enhance the character of the whole crop rice silage and thus offer an excellent fermentation end product with a high protein and low NDF content [50], and applying these additives to raise the quality of CSP produces successful outcomes.

### 4.3. The Fermentation Characteristics of Cassava Pulp Fermented with LcTH14, Urea, and Molasses after Fermentation

Neumann et al. [45] verified that silages treated with urea present higher pH values. Similar to Kang et al. [51], the pH of silage increased with urea supplementation. However, McDonald et al. [52] reported the pH level was decreased by the addition of 2% of molasses, and Yuan et al. [53] found a critical value that indicates silage is appropriate and the pH declines below 4.0 after 2 days of ensiling. Different fermentation days resulted in a slight increase in pH on the 14th day of the fermentation of CSP, which remained steady until the 21st day. Furthermore, Yitbarek and Tamir [54] cited that anhydrous ammonia-treated corn silage typically has a pH that is 0.1 to 0.2 units higher when the silo’s fermentation process has been completed. In addition, urea is an alkalinizing additive, therefore adding it to the silage caused the pH to increase [55].

LcTH14 was the homofermentative type of LAB; it has a lactic capacity, and could produce more LA, which is more effective in improving silage quality [16]. Homofermentative inoculants include bacteria such as *Lactobacillus plantarum*, *Pedi coccus*, and *Lactococcus* species. They support ensure fermentation, which produces LA and rapidly lowers the pH to 4, inhibiting the degradation of the crop’s protein and sugar [54,56]. During ensiling, because LA produced by lactic acid bacteria (LAB) is normally the acid present in the highest concentration in silages and is about 10–12 times stronger than any of the other main acids (such as AA and PA) present in silages, it is responsible for the pH decline that occurs during fermentation [57]. Yunus et al. [38] found that LA production was increased by the addition of molasses but decreased by urea. In this experiment, the pH increased as the fermentation duration climbed, and the LA content increased as well. The initial pH of the treated silages improved as intended; the NH_3_ released from urea buffered the silage [58]. According to Shirley et al. [59], this is due to the buffering function of urea, which mitigates the silage’s extreme acidity. This helps keep the silage from becoming too acidic [41]. Moreover, urea treatment increases the pH range in silages, which has facilitated the growth of various LAB types [41]. This may be the reason that pH can be increased along with LA in this study. The period of fermentation days increased as well as the concentration of LA and AA, as was observed in this study. Fermentation actually occurred in this experiment after 21 days. According to Wang and Nishino’s research [60], TMR silage’s LA and AA concentrations increased when ensiling time was prolonged.

*Propionibacterium* spp., which may ferment lactate to propionate, have been identified sometimes in low pH silages, according to Woolford [61], who also noted that the presence of sugar had no effect on this fermentation. *Propionibacterium* that convert glucose and LA to AA and PA have been found in silages, but it is doubtful that natural populations can flourish in most silages [56]. Lactate is the most essential carbon source for *Propionibacterium* spp. in habitats with high LA concentrations (e.g., silages) [62]. Studies have showed that the concentration of PA increases as the number of days of fermentation increases. The butyric acid (BA) ranged from 1.14 to 1.71 g/kg DM during the entire fermentation process. The BA ranged from 1.14 to 1.71 g/kg DM and was higher on 21 days than on 14 days of fermentation. Though the silage has undergone clostridial fermentation, which is one of the worst fermentations, it will have a high concentration of BA (more than 5 g/kg DM) [63]. 

Ammonia-N production is affected by the CP breakdown of silage, where the fraction of proteolysis is shown by the generation of ammonia-N, and well-preserved silages should contain no more than 100 g NH_3_-N/kg total nitrogen [52]. In the present study, the higher ammonia-N content was achieved by the addition of L×U and U×M to CSP fermentation and the high (ranging from 1.02 to 54.62 g/kg DM) ammonia-N by urea addition. Similarly, Fang et al. [55] reported that urea-treated rice-straw silage resulted in high ammonia-N content and total nitrogen from urea, and Santos et al. [40] discovered that as the levels of urea addition were enhanced, the ammoniacal nitrogen increased linearly. Urea will be hydrolyzed to ammonia-N during the period of fermentation when applied to forage crops by urease enzymes [64]. Garcia Alvaro [65] reported that ammonia-N breakpoints are between 1.7 to 109.4 g/kg. In this experiment, the ammonia-N values of CSP fermented with non-urea and urea-containing treatments were 1.02–2.23 g/kg DM and 32.61–54.62 g/kg DM, respectively. The ammonia-N value of the fermented CSP containing urea was found to be lower than the breakpoints. Moreover, because the pH is below 4, the fermentation process was accomplished. Frequently, silage with high levels of ammonia may also contain considerable amounts of other undesirable end products, such as amines, which could reduce animal performance [66]. However, the blood or milk urea nitrogen can be used as an indicator of excess ruminal-degraded protein (RDP) if the excessive ammonia contributes to an abundance of RDP [66].

### 4.4. Microbiology and Aerobic Stability of Cassava Pulp Treated with LcTH14, Urea, and Molasses after 21 Days of Fermentation

Ensilage, also known as ensiling, is a method of conserving forage for use as animal feed under anaerobic conditions. The purpose is to prevent oxygen contact with forage for a period of time due to yeast and mold that will develop as a result of aerobic microbial activity, causing the material to degrade into an unusable, unpalatable, and always poisonous product [53]. Silage additives have been utilized in wet silages (25 to 30 g DM/kg) to enhance fermentation [64]. In this study, LAB inoculation resulted in a higher LAB population in the fermented CSP than in the uninoculated group. In addition, the LAB population of CSP treated with LM additive was the highest, followed by LUM, L, and other, respectively. The AB population in the CSP treated with the addition had lower populations than the control. A microbial inoculant was added to the silage to introduce homofermentative lactic acid bacteria (hoLAB), which can grow quickly and take control of the fermentation, leading to higher-quality silage [54]. As microbial activity is inhibited, any path’s conservation efficiency should have included acid production and energy losses; the path with the least energy losses, homo-fermentative LAB, creates two lactic acid (pKa 3.86) molecules [67]. Limin Kung Jr. reported that lactic acid fermentations lost the least amount of crop dry matter and energy during storage [66]. Cherdthong et al. [68] found LAB inoculate decreased the AB and yeast populations of ensilaged rice straw. In accordance with this, Keawpila et al. [33] reported the numbers of yeast were reduced by LAB inoculation in silage. Accordingly, the ensiling additive with *L. casie* TH14, cellulase, and molasses in combination has resulted in the promotion of the best qualities of rice straw and sugarcane bagasse, respectively [68,69]. In addition, the combination of LAB and enzymes resulted in a faster pH decrease, which enhanced both the fermentation quality and the aerobic stability of tomato pomace and pumpkin silage [70]. 

While silage is exposed to the air, the ambient temperature and the silage temperature are observed. Aerobic stability is the period of time until the silage effectively exhibits signs of heating, or when the silage’s condition is 2 °C or higher above ambient. As a result, the microorganisms oxidize acids and water-soluble carbohydrates to produce water and carbon dioxide, which increases the temperature of the silage above the ambient air [30]. Pahlow et al. [71] found that all 264 legume silages were stable after being exposed to air for at least 96 h (4 days), and 89% were stable after 168 h (7 days). Moreover, in thirteen experiments including three different plants (whole-crop maize, wheat, and barley), inoculation with *L. plantarum* with sorbate resulted in an overall increase in mean aerobic stability of 77 h for untreated silage and 153 h for treated silage [57,72]. Moreover, aerobic stability was clearly improved by inoculation (it regulates silage heating after 26 h while treated silages were maintained cool for over 400 h in silage) [54]. In this investigation, aerobic stability remained consistent for more than 120 h after silo opening, indicating that CSP treated with *Lactobacillus casei* TH14, urea, and molasses could enhance silage quality.

## 5. Conclusions

According to the current findings, CSP is satisfactorily preserved when treated with a combination of *Lactobacillus casei* TH14, urea, and molasses. The combination of treatments causes CSP to regain DM, have low levels of aerobic bacteria, and have greater amounts of LAB. We recommended fermenting CSPs for more than 21 days in this experiment since it produced higher CP and LA content than other procedures. However, further research on in vitro methods for ruminant feed should be assessed.

## Figures and Tables

**Table 1 vetsci-09-00617-t001:** Microbial counts, chemical composition, and pH of wet cassava pulp before fermentation.

Item ^1^	Cassava Pulp
Microbial counts (cfu/g FM)	
LAB	8.8 × 10^4^
Coliform bacteria	ND
Aerobic bacteria	1.91 × 10^8^
Yeasts	1.98 × 10^7^
Molds	ND
Chemical composition (g/kg DM)	
DM (g/kg)	165.50
OM	973.60
CP	25.94
EE	2.53
NDF	438.87
ADF	246.79
pH	3.96

^1^ cfu = colony forming unit, g = gram, FM = fresh matter, LAB = Lactic acid bacteria, ND = not-detected, DM = dry matter, kg = kilogram, OM = organic dry matter, CP = crude protein, EE = ether extract, NDF = neutral detergent fiber, ADF = acid detergent fiber.

**Table 2 vetsci-09-00617-t002:** Effect of cassava pulp by treated with *Lactobacillus casei* TH14, urea, and molasses on the chemical composition of silage.

Additive ^1^	DM (g/kg)	OM	CP	EE	NDF	ADF
		(g/kg DM)				
CON	166.10 ^d^	917.25	22.10	4.52	394.37	222.03
M	172.19 ^bc^	919.09	22.55	4.81	359.56	207.80
U	175.19 ^ab^	914.12	138.55	3.72	346.32	222.94
UM	177.62 ^a^	915.80	129.64	3.32	335.67	220.54
L	166.69 ^d^	915.31	21.34	4.20	353.90	198.84
LM	169.62 ^cd^	919.94	23.46	4.09	334.71	216.77
LU	168.54 ^cd^	914.90	154.69	3.69	347.00	214.96
LUM	177.23 ^a^	915.87	132.91	2.93	339.29	200.53
SEM	1.35	2.91	2.86	0.57	9.60	8.25
*p*-value Interaction						
L × U × M	<0.05	0.68	0.09	0.99	0.65	0.08
L×U	0.21	0.82	<0.05	0.71	<0.05	0.57
L × M	0.43	0.81	0.19	0.63	0.51	0.40
U × M	0.59	0.65	<0.01	0.42	0.21	0.40
Main effect						
L	<0.05	0.98	0.03	0.38	0.04	0.10
U	<0.01	0.21	<0.01	<0.05	<0.05	0.57
M	<0.01	0.29	<0.01	0.55	<0.05	0.58

^1^ CON = no additive, M = treated with molasses 4% DM, U = treated with urea 4% DM, UM = treated with urea 4% DM and molasses 4% DM, L = treated with LcTH14 10^5^ cfu/kg FM, LM = treated with LcTH14 10^5^ cfu/kg FM and molasses 4% DM, LU = treated with LcTH14 10^5^ cfu/kg FM and urea 4% DM, and LUM = treated with LcTH14 10^5^ cfu/kg FM, urea 4% DM, and molasses 4% DM, SD = standard deviation, DM = dry matter, g = gram, kg = kilogram, OM = organic dry matter, CP = crude protein, ^a–d^ Means within columns with difference superscript differ at *p* < 0.05 and *p* < 0.01.

**Table 3 vetsci-09-00617-t003:** Effect of cassava pulp by treated with *Lactobacillus casei* TH14, molasses, and urea on fermentation characteristics after fermentation.

Additive ^1^	pH	LA	AA	PA	BA	Ammonia-N
		(g/kg DM)				
CON	3.47	45.29	27.41	13.57	1.56	2.23
M	3.50	45.29	23.93	11.66	1.35	1.02
U	3.61	50.97	26.24	11.18	1.43	51.66
UM	3.49	48.34	26.42	12.30	1.54	32.61
L	3.47	61.14	24.69	11.66	1.50	1.07
LM	3.47	85.77	25.49	11.61	0.88	1.54
LU	3.59	54.53	26.77	12.54	1.52	54.62
LUM	3.49	72.40	27.66	13.52	1.66	38.38
SEM	0.01	9.65	2.86	0.57	9.60	8.25
*p*-value Interaction						
L × U × M	0.13	0.88	0.36	0.24	0.44	0.73
L × U	0.70	0.31	0.45	0.01	0.20	<0.05
L × M	0.56	0.12	0.21	0.30	0.51	0.20
U × M	<0.01	0.74	0.34	<0.05	0.07	<0.01
Main effect						
L	0.05	<0.01	0.87	0.70	0.58	<0.05
U	<0.01	0.69	0.16	0.53	0.13	<0.01
M	<0.01	0.17	0.68	0.93	0.31	<0.01

^1^ CON = no additive, M = treated with molasses 4% DM, U = treated with urea 4% DM, UM = treated with urea 4% DM and molasses 4% DM, L = treated with LcTH14 105 cfu/kg FM, LM = treated with LcTH14 10^5^ cfu/kg FM and molasses 4% DM, LU = treated with LcTH14 10^5^ cfu/kg FM and urea 4% DM, and LUM = treated with LcTH14 10^5^ cfu/kg FM, urea 4% DM, and molasses 4% DM, DM = dry matter, ammonia-N = ammonia nitrogen, LA = lactic acid, AA = acetic acid, PA = propionic acid, BA = butyric acid.

**Table 4 vetsci-09-00617-t004:** Effect of cassava pulp by treated with *Lactobacillus casei* TH14 with urea and molasses on microbial population after 21 days of fermentation.

Additive ^1^	Microbial Counts		
	LAB	Aerobic Bacteria	Yeasts
	10^6^ cfu/g FM	10^5^ cfu/g FM	10^4^ cfu/g FM
CON	0.22 ^d^	9.00 ^a^	1.00
M	0.74 ^d^	3.60 ^c^	ND
U	0.48 ^d^	1.77 ^d^	1.33
UM	0.62 ^d^	0.82 ^d^	0.33
L	2.14 ^c^	5.40 ^b^	0.50
LM	9.10 ^a^	4.60 ^cb^	2.50
LU	1.38 ^cd^	0.67 ^d^	0.50
LUM	3.87 ^b^	0.29 ^d^	1.00
SEM	0.37	0.48	0.74
*p*-value Interaction			
L × U × M	<0.01	<0.05	0.65
L × U	<0.01	0.55	0.22
L × M	<0.01	<0.01	0.08
U × M	<0.01	<0.01	0.67
Main effect			
L	<0.01	<0.01	0.28
U	<0.01	<0.01	0.55
M	<0.01	<0.01	0.96

^1^ CON = no additive, M = treated with molasses 4% DM, U = treated with urea 4% DM, UM = treated with urea 4% DM and molasses 4% DM, L = treated with LcTH14 10^5^ cfu/kg FM, LM = treated with LcTH14 10^5^ cfu/kg FM and molasses 4% DM, LU = treated with LcTH14 10^5^ cfu/kg FM and urea 4% DM, and LUM = treated with LcTH14 10^5^ cfu/kg FM, urea 4% DM, and molasses 4% DM, cfu = colony forming unit, g = gram, FM = fresh matter, LAB = lactic acid bacteria, ND = not detected (<10^2^ cfu/g FM), ^a–d^ Means within columns with difference superscript differ at *p* < 0.01.

## Data Availability

Data available in a publicly accessible repository.

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
