# Peer review of "The Chemical Composition, Fermentation End-Product of Silage, and Aerobic Stability of Cassava Pulp Fermented with Lactobacillus casei TH14 and Additives"

_vetsci, 2022, doi:10.3390/vetsci9110617_

Round 1

Reviewer 1 Report

This manuscript is within the scope of the Nutritional and Metabolic Diseases in Veterinary Medicine section of the Veterinary Science journal, but it had some serious issues that have to explain.

1.  What is the novelty of this study? The lactic acid bacteria (LAB) inoculation and molasses supplementation are already used in many previous studies and approved its benefit to improve the silage quality. The agriculture byproduct or food processing byproduct had also been reported to improve storability through the ensilage treatment. If the LAB strain applied in this study had some special characteristics, it should be described in the introduction section. The major aim of these treatments is to improve the storage stability or improve the nutrition value of the CSP?

2.  How did the author set the molasses and urea supplementation levels? The chemical composition of CSP showed that CSP contained a high level of nitrogen-free extract, but no detailed chemical assay result about water soluble carbohydrate (WSC) or starch was shown in the present manuscript. If the WSC is rich in CSP, how about the role of molasses supplementation in the present study?

3.  According to the data in Table 1, the pH of CSP before fermentation was 3.96 and it also had pretty high LAB counts (8.8x104 cfu/g FM). However, the L. casei TH14 inoculation level was 105 cfu/g FM and it is close to the LAB counts in CSP. How to confirm the effect of L. casei TH14 on the fermentation process?

4.  Both Table 2 and Table 3 showed the assay data on different days after fermentation. However, the assay results are presented as the average value of each assay item. The different additive should result in different performance during the fermentation process, the average value of each treatment at the same day only provide very limited information. The fermentation product or chemical composition change of individual treatment at different fermentation days is more suitable for explaining the result.

5.  In the discussion section, the related issue of dry matter loss during fermentation and energy conversion seems weak in the present manuscript. In L410-419, the author cited many previous research data about forage silage. The discussion about LAB inoculation into byproducts or food byproducts should be shown in the discussion section.

6.  In this manuscript, Table 3 only showed the fermentation characteristics after fermentation but no clear information let us know the detailed assay result came from which sampling day. The chemical composition, lactic acid, SCFA, and ammonia-N data of each treatment at fermentation day 0 were also unavailable.   

7.  In conclusion, the author indicated the CPS fermentation period should be over 21 days to produce higher CP. However, the CP increase in Table 2 of this study may result from the urea supplementation. On the other hand, the high NH3 coming from urea additives also make a problem with odor in the product.

8.  Other comments about this study:

L19: The FM (is it mean "fresh matter"?) should be defined in the abstract and material section.

L140-143: How the silage sampling for microbial analysis should be described simply.

L152-154: How about the packing density of the silage for aerobic stability analysis? The packing density could affect the stability after the silage is exposed to air.

L438-440: According to the results, which treatment is recommended for CPS fermentation? Why? 

Author Response

Response to Reviewer 1

This manuscript is within the scope of the Nutritional and Metabolic Diseases in Veterinary Medicine section of the Veterinary Science journal, but it had some serious issues that have to explain.

Response: We would like to sincerely thank Reviewer 1, who provided a positive comment and useful suggestion to improve this manuscript. In the present revised version, we have tried our best to modify as comments and suggestions. Please see the manuscript.

1. What is the novelty of this study? The lactic acid bacteria (LAB) inoculation and molasses supplementation are already used in many previous studies and approved its benefit to improve the silage quality. The agriculture byproduct or food processing byproduct had also been reported to improve storability through the ensilage treatment. If the LAB strain applied in this study had some special characteristics, it should be described in the introduction section. The major aim of these treatments is to improve the storage stability or improve the nutrition value of the CSP?

Response: Thank you for suggesting this very helpful advice for this experiment. We'd like to inform you of the following:

1) The novelty of this work is that we wish to introduce the novel lactic acid bacteria (Lactobacillus casei strain TH14) discovered by Pholsen et al. (2016, Anim. Sci. J. 87, 1202-1211; a previous department staff) in order to improve the nutritional value of CSP in combination with additives. In addition, a review of the literature reveals that Lactobacillus casei strain TH14 inoculation with CSP has not been previously documented. Thai farmers have utilized CSP as ruminant feed extensively, however the utilization is limited, as seen by the studies we present in Introduction. Thus, we endeavored to explore various strategies of enhancement with the novel Lactobacillus casei strain TH14 that could be advantageous to farmers. The following are the unique properties of the LAB strain used in this study: Pholsen et al. present evidence comparing homofermentative types of LAB, such as Lactobacillus casei strain TH14, Lactobacillus plantarum for 3 strains, and Lactobacillus rhamnosus. They discovered that strain TH14 has a high lactic acid production capacity and can produce more lactic acid, confirming its applicability as a possible silage inoculant and demonstrating that this strain was more successful than inoculants or other strains in enhancing silage quality. Please see lines 76-81.

2) Present work aims to improve the nutritional content of CSP by adding an addition to the silage, while silage quality is also evaluated to verify that CSP can be effectively preserved using our method. Thus, as we set up the aim that this study was to evaluate the effects of fermented cassava pulp (CSP) with L. casei TH14 (LcTH14), urea, and molasses on chemical composition, the fer-mentation end-product of silage, microbial population, and aerobic stability. 

2. How did the author set the molasses and urea supplementation levels? The chemical composition of CSP showed that CSP contained a high level of nitrogen-free extract, but no detailed chemical assay result about water soluble carbohydrate (WSC) or starch was shown in the present manuscript. If the WSC is rich in CSP, how about the role of molasses supplementation in the present study?

Response: Thank you very much. We'd like to inform you of the following:

1) For the molasses and urea supplementation levels, we refer to the experimental work of Norrapoke et al. (2008; J. Appl. Anim. Res, 46, 242-247), who investigated the fermentation of cassava pulp using urea and molasses at a level of 0 to 6% dry matter. They discovered that at 4% urea and 4%molasses could improve gas production kinetics, in vitro digestibility, and we have previously cited it in the manuscript. Please see lines 104-108.

2) In the present work, we did not determine NFE or WSC because we know that CSP has very little of these components. According to Tonsing et al. (2008; Kasetsart J. Nat. Sci. 42: 627–631), CSP was composed of 66% NFE and 13% high crude fiber. Thus, we anticipated that extra molasses could serve as a carbon source for LAB, speed up the fermentation process, and improve silage quality.

3. According to the data in Table 1, the pH of CSP before fermentation was 3.96 and it also had pretty high LAB counts (8.8x104 cfu/g FM). However, the L. casei TH14 inoculation level was 105 cfu/g FM and it is close to the LAB counts in CSP. How to confirm the effect of L. casei TH14 on the fermentation process?

Response: Thank you very much. We can confirm that L. casei TH14 could influence the fermentation process. As evidenced by the average pH value after fermentation with L. casei TH14 (Table 3), pH decreased to 3.51, which is 0.44 units lower than CSP prior to fermentation. In addition, when other fermentation characteristics are observed, the L. casei TH14 inoculant can be improved more so than previously CSP.

4. Both Table 2 and Table 3 showed the assay data on different days after fermentation. However, the assay results are presented as the average value of each assay item. The different additive should result in different performance during the fermentation process, the average value of each treatment at the same day only provide very limited information. The fermentation product or chemical composition change of individual treatment at different fermentation days is more suitable for explaining the result.

Response: Thank you for your suggestion. Based on your suggestions, we have added a chart of the average value of each treatment at different days as well as a mean of each day of fermentation, to help the reader understand how the day of fermentation affects the results in the appendix. This wouldn't disrupt the flow of the main text. Please see Figure 1. Furthermore, we have included a reference to this part in the manuscript. Please see a section for Results and a section for Discussion.

5. In the discussion section, the related issue of dry matter loss during fermentation and energy conversion seems weak in the present manuscript. In L410-419, the author cited many previous research data about forage silage. The discussion about LAB inoculation into byproducts or food byproducts should be shown in the discussion section.

Response: Thank you for providing this helpful information. We incorporated more important information about the dry matter loss as “This study's DM content is consistent with the findings of So et al. [21], who discovered that fermenting sugarcane bagasse with Lactobacillus casei TH14 combined with cellulase, and molasses did not reduce DM loss. In addition, Cherdthong et al. [17] found no difference in DM loss after ensiled rice straw was treated with Lactobacillus casei TH14, molasses, and cellulase enzymes”. Please see at lines 317-321.

6. In this manuscript, Table 3 only showed the fermentation characteristics after fermentation but no clear information let us know the detailed assay result came from which sampling day. The chemical composition, lactic acid, SCFA, and ammonia-N data of each treatment at fermentation day 0 were also unavailable.   

Response: Thank you for your suggestion. We'd like to inform you of the following:

1) To help the reader understand how the day of fermentation impacts the results in the appendix, we have included a chart of the average value of each treatment at different days, as well as a mean of each day of fermentation. This would not interfere with the flow of the main content. Please take a look at Figure 1.

2) Because no fermentation process has occurred from each addition on day 0 of fermentation. The chemical composition, lactic acid, SCFA, and ammonia-N, would be unaltered by the fermentation process. We believe that the data in this item is superfluous, and day 0 was not recognized.

7. In conclusion, the author indicated the CPS fermentation period should be over 21 days to produce higher CP. However, the CP increase in Table 2 of this study may result from the urea supplementation. On the other hand, the high NH3 coming from urea additives also make a problem with odor in the product.

Response: Thank you for your suggestion. We agreed that greater CP over 21 days could be caused by urea supplementation. However, we anticipated that protein from urea would be advantageous when CSP silage was fed to ruminants throughout the fermentation process. Furthermore, NH3 in CSP silage after 21 days of fermentation may have more opportunity to function as an alkali and breakdown fiber content in CSP, making it easier to digest and utilize by ruminants. Based on our actual experience and testing, we can certify that NH3 derived from urea additions did not cause much odor in the product and that NH3 may be quickly lost when the silo is opened.

  1. Other comments about this study:

L19: The FM (is it mean "fresh matter"?) should be defined in the abstract and material section.

Response: Thank you very much. We have fixed it. Please see line 19.

L140-143: How the silage sampling for microbial analysis should be described simply.

Response: Thank you very much. We have added the protocol for microbial sampling. Please see lines 154-170.

L152-154: How about the packing density of the silage for aerobic stability analysis? The packing density could affect the stability after the silage is exposed to air.

Response: Because this experiment was conducted for laboratory-scale experiments, only 5000 grams of the materials were divided into 350-gram bags and vacuum-sealed by Brother (Zhejiang Brother Packing Machinery Co., Ltd., Zhejiang, China) in order to achieve a packing density at room temperature. This was done by Zhejiang Brother Packing Machinery Co., Ltd., which is located in Zhejiang, China. After opening the bags, a speedy analysis was performed. The bags had been tightly packaged. As a result, we were able to establish that the packing density would not have an effect on the silage's stability after it had been exposed to air.

L438-440: According to the results, which treatment is recommended for CPS fermentation? Why? 

Response: Thank you very much. We recommended CPS fermentation with L. casei TH14, urea, and molasses since the findings had a positive influence on the chemical composition and overall quality of the fermentation.

1) Because just urea was added, the pH of the fermentation increased and the quality of the silage decreased.

2) When adding molasses or L. casei TH14 alone, the protein content will be low, but the pH and other parameters will be appropriate.

Therefore, we expected that the aforementioned would clearly guide the reader through the data presented in the manuscript and that the combination of three chemicals in CSP would be recommended for ruminant feed production.

Thanks again for your valuable comments and suggestion.

Reviewer 2 Report

The manuscript is interesting and the topic fits within the journal. There some deficiencies which raised my concern.

Throughout the manuscript (abstract, results section) the authors use significantly and the p value at the same time. This is not necessary please use only significant or only p-value. Using both is a repetition which is not needed.

Line 54-56 Ogunbode et al (Ref No 9) did not show any thing in Thai beef cattle…this reference is not correct

Line 56-58 you should indicate how much cassava was introduced in the diet and that the protein content was reduced to below 10 %. Therefore it is not surprising that urinary and retained N decreased…please give all necessary description of cited references so that the reader can follow your description wihot looking in to the cited paper…

Line 66   please xplain fort he reader shortly how SC improves Quality of byproducts

Line 70 -85 Is it really a good idea to compare eniling of rice straw and sugarcane bagasse which are high in low fermentable carbohydrates with cassava pulp which still has high amount of fermentable substrate?

Line 189- 193 is it really necessary to repeat the results from Table1 again in the text. Please delete one: Table or text!!

Please make sure that tables are not divided over two pages, makes it hard to read. TAbles should always be on one page

Table 7 Please delete columns with aerobic stability, coliform bacteria and molds from Table 7. Values are always the  same or not detectable. It is sufficient to write this in one sentence in the text.

In the discussion,there is a lot repetition of the results at the beginning of each paragraph. This should be avoided and disattracts from reading

Line 268  please do not use the wording growth promoter for starch!! It is a highly fermentable carbohydrate…how much starch is left in your cassava pulp please analyze the starch content!

Line 276 enhanced with a high protein content…please give a number! High is very unspecific

The discussion is way too long and not well structured, it should be shortened and only the important results should be discussed

Author Response

Response to Reviewer 2

The manuscript is interesting and the topic fits within the journal. There some deficiencies which raised my concern.

Response: We would like to sincerely thank Reviewer 2, who provided a positive comment and useful suggestion to improve this manuscript. In the present revised version, we have tried our best to modify it as comments and suggestions. Please see the manuscript.

Throughout the manuscript (abstract, results section) the authors use significantly and the p value at the same time. This is not necessary please use only significant or only p-value. Using both is a repetition which is not needed.

Response: Response:  Thank you very much. We have fixed it. Please see throughout the manuscript.

Line 54-56 Ogunbode et al (Ref No 9) did not show any thing in Thai beef cattle…this reference is not correct

Response: Thank you very much. We have fixed it and referred to the correct citation as “According to Kongphitee et al. [9], the inclusion of 50% CSP to the total mixed ration (TMR) could reduce the intake of acid detergent fiber (ADF) by 54% in Thai native beef cattle.” Please see lines 54-56.

Line 56-58 you should indicate how much cassava was introduced in the diet and that the protein content was reduced to below 10 %. Therefore it is not surprising that urinary and retained N decreased…please give all necessary description of cited references so that the reader can follow your description wihot looking in to the cited paper…

Response: Thank you very much. We have added a necessary description of cited references following “Additionally, Keaokliang et al. [10] illustrated that Thai native beef cattle fed with a TMR diet (9.6% crude protein; CP) containing 29.8% CSP could reduce protein digestion, urinary nitrogen, retained nitrogen, and the urine to gross energy ratio in comparison to the control group, which was fed a TMR diet with 12.6% crude protein and no CSP”. Please see lines 56-60.

Line 66   please xplain fort he reader shortly how SC improves Quality of byproducts

Response: Thank you very much. We have provided more explanation as “This might be because S. cerevisiae has a high protein content and is abundant in the essential amino acids that animals require [13]. Additionally, the capability of yeast cells to scavenge oxygen is one of the primary elements that may be one of the primary factors that may justify the beneficial effect that live yeasts have on the quality of silage as well as fiber-degrading bacteria in the rumen [12,14]. In a similar fashion, Suntara et al. [12] found that certain strains of yeast were able to produce cellulolytic enzymes, which were capable of degrading the fibers present in plant materials.” Please see lines 60-76.

Line 70 -85 Is it really a good idea to compare eniling of rice straw and sugarcane bagasse which are high in low fermentable carbohydrates with cassava pulp which still has high amount of fermentable substrate?

Response: Thank you very much. Even while cassava pulp includes a greater quantity of fermentable substrates than rice straw, some fermentable substrates may not be adequate for LAB. In an effort to increase the availability of fermentable sources for L. casei TH14, sugarcane bagasse may be considered for inclusion. In addition, the combination of fermentation additives to improve agricultural and industrial waste is another target that may have a more practical application for farmers and be recommended if the research is successful.

Line 189- 193 is it really necessary to repeat the results from Table1 again in the text. Please delete one: Table or text!!

Response:  Thank you very much. We have modified it. Please see lines 207-209.

Please make sure that tables are not divided over two pages, makes it hard to read. TAbles should always be on one page

Response: Thank you for your suggestion. We were able to eliminate it entirely, but a few remnants remained as follows: a part of Table 3 was moved to the following page, as was the description under Table 4. Please see the text.

Table 7 Please delete columns with aerobic stability, coliform bacteria and molds from Table 7. Values are always the same or not detectable. It is sufficient to write this in one sentence in the text.

Response:  Thank you very much for your nice recommendation. However, we believe that these data should stay in Table 4 so that the reader can gain a better understanding of each condition and treatment in terms of aerobic stability, coliform bacteria, and mold analysis. Some researchers prefer to swiftly scan the Tables rather than the text. In addition, several of the previously published papers on silage quality contain similar findings, which is acceptable.

In the discussion, there is a lot repetition of the results at the beginning of each paragraph. This should be avoided and disattracts from reading

Response: Thank you very much. We have fixed it. Please follow the discussion section

Line 268 please do not use the wording growth promoter for starch!! It is a highly fermentable carbohydrate…how much starch is left in your cassava pulp please analyze the starch content!

Response: Thank you very much. We have fixed it as “Additionally, a high population of yeast and AB was found in unfermented cassava pulp [33], which may be due to their ability to utilize the carbohydrate that exists as a partial substrate for microbe growth [34].”. Please see lines 283-286.

Line 276 enhanced with a high protein content…please give a number! High is very unspecific

Response: Thank you very much. We have fixed it as “It has been shown that fermented CSP increases CP by 12.7% to 26.4% [12,13,20], which ultimately results in an increase in the nutritional content of animal diets [6]”. Please follow line 292-294.

The discussion is way too long and not well structured, it should be shortened and only the important results should be discussed

Response: Thank you so much for your feedback. We have done our utmost to eliminate superfluous conversations, but some remain. Please refer to the entirety of the Discussion section.

Thanks again for your valuable comments and suggestion.

Round 2

Reviewer 1 Report

The descript and the figure data had improved and more clear in the present version.

Only one suggestion for the revised manuscript: The resolution of Figure A1 could be improved and the size could be magnified.

Author Response

Response to Reviewer 1

The descript and the figure data had improved and more clear in the present version.

Only one suggestion for the revised manuscript: The resolution of Figure A1 could be improved and the size could be magnified.

Response: Thank you for your suggestion.  We have made this improvement by separating the figures into two different ones so that we can enhance both the resolution and the size. Please refer to the manuscript for more information.

Thanks again for your valuable comments and suggestion.

Reviewer 2 Report

Line 73-74 reference 14 is on dairy heifers why do they need essential amino acids from S. cerevisiae  ? and how much Saccharomyces do you add ? that barely changes the protein content…please rewrite

Table 4 and discussion line 478 - 496

I don’t see any reason why the aerobic stability,coliform bacteria and molds column should stay in Table 4, this does not bring any further message. Just simply mention this in the text. In addition there is no difference obtained for these values and no statistics. In this study even the control without any additive has the same aerobic stabilty. I would expect that you discuss reasons why aerobic stability is not changed due to the additives. Why discuss nonsignificant results at all.

Author Response

Response to Reviewer 2

Line 73-74 reference 14 is on dairy heifers why do they need essential amino acids from S. cerevisiae  ? and how much Saccharomyces do you add ? that barely changes the protein content…please rewrite

Response: Thank you for your comments. In order to make it clear, we have changed citations related to the content and added some explanation supporting yeast utilization as “This could be because S. cerevisiae has a high protein content and is rich in the essential amino acids animals require. Phesatcha et al. [15] showed that adding 15 g/d of S. cerevisiae to fattening cattle enhanced nutrient digestibility and growth performance.” Pleases see in Line 69-71.

Table 4 and discussion line 478 - 496

I don’t see any reason why the aerobic stability,coliform bacteria and molds column should stay in Table 4, this does not bring any further message. Just simply mention this in the text. In addition there is no difference obtained for these values and no statistics. In this study even the control without any additive has the same aerobic stabilty. I would expect that you discuss reasons why aerobic stability is not changed due to the additives. Why discuss nonsignificant results at all.

Response: Thank you for your comments. Now, we have removed data of aerobic stability, coliform bacteria and molds Table 4. The discussion on aerobic stability was provided in Line 434-448. Please see in manuscript.

Thanks again for your valuable comments and suggestion.
